# Escaping Mode Collapse in LLM Generation via Geometric Regulation

**Xin Du** [1] [2]   **Kumiko Tanaka-Ishii** [3]

## Abstract

Mode collapse is a persistent challenge in generative modeling and manifests in autoregressive text generation as behaviors ranging from explicit looping to gradual loss of diversity and premature trajectory convergence. We take a dynamical-systems view and reinterpret mode collapse as reduced state-space accessibility caused by *geometric collapse*: during generation, the model's internal trajectory becomes confined to a low-dimensional region of its representation space. This implies mode collapse is not purely a token-level phenomenon and cannot be reliably mitigated by symbolic constraints or probability-only decoding heuristics. Guided by this perspective, we propose *Reinforced Mode Regulation* (RMR), a lightweight, online state-space intervention that regulates dominant self-reinforcing directions in the Transformer value cache (implemented as low-rank damping). Across multiple large language models, RMR substantially reduces mode collapse and enables stable, high-quality generation at extremely low entropy rates (down to 0.8 nats/step), whereas standard decoding typically collapses near 2.0 nats/step.

## 1. Introduction

Large language models (LLMs) can generate remarkably fluent text, yet long-horizon decoding remains fragile: small local biases can accumulate into macroscopic failure modes. A generation process may gradually lose diversity and become repetitive, bland, or abruptly trapped in self-reinforcing patterns. We refer to this family of long-range failures as *mode collapse*. A key challenge is that mode collapse is hard to

define locally or predict from individual next-token probabilities, even though its consequences dominate downstream usability in long-form generation.

Much of the decoding literature addresses these failures by modifying the next-token distribution—through truncation, penalties, or sampling variants. While often effective in practice, such methods operate at the symbolic level and are inherently local: they do not directly characterize the evolving *internal state* that gives rise to long-range behaviors. As a result, they can mitigate symptoms but offer limited insight into why collapse emerges systematically in certain regimes, nor do they provide a principled handle for controlling long-horizon dynamics.

We propose a geometric, dynamical-systems view of this phenomenon. We model autoregressive decoding as a stochastic trajectory in a high-dimensional state space (e.g., the Transformer KV cache). In this picture, mode collapse is associated with *geometric collapse*: the internal trajectory loses *state-space accessibility* and becomes confined to a low-dimensional metastable region that is difficult to escape. This reframes the central question from "which tokens repeat" to "why does the internal dynamics become effectively low-dimensional," and it suggests that purely symbolic constraints or probability-only heuristics may be insufficient in the regimes where collapse emerges.

We empirically support this view by measuring the *correlation dimension* of LLM generation trajectories. Correlation dimension is a trajectory-wise fractal dimension that quantifies the dynamically active degrees of freedom. Thus, it provides a direct geometric measure of accessibility. Across real LLM decoding, it consistently reflects mode collapse on the text surface (including explicit looping and increasing repetition), while remaining conceptually distinct from token-level proxies such as entropy or Distinct-$n$.

Guided by this diagnosis, we seek to prevent mode collapse by intervening in the model's *state space* rather than only reshaping the next-token distribution. We propose *Reinforced Mode Regulation* (RMR), a lightweight inference-time intervention that selectively attenuates the dominant self-reinforcing directions in the Transformer value cache. RMR identifies a small low-rank subspace exhibiting unusually strong temporal persistence by solving a generalized eigenvalue problem with bounded spectrum. We then apply

[1]Department of Communications and Computer Engineering, Waseda University, Tokyo, Japan [2]Shanghai Research Institute for Intelligent Autonomous Systems, Tongji University, Shanghai, China [3]Department of Computer Science and Engineering, Waseda University, Tokyo, Japan. Correspondence to: Kumiko Tanaka-Ishii <kumiko@waseda.jp>.

*Proceedings of the 43rd International Conference on Machine Learning*, Seoul, South Korea. PMLR 306, 2026. Copyright 2026 by the author(s).

principled thresholding to target only the most persistent directions, and implements the regulation as a low-rank update to the value cache.

Across multiple LLMs and decoding settings, RMR substantially reduces mode-collapse incidence, including in very low-randomness regimes where standard decoding nearly always collapses. For instance, RMR improves non-collapse rates from 8% to 56% at temperature 0.7 and from 5% to 33% at an entropy target of 1.0. This demonstrates that geometric, state-space regulation can stabilize long-horizon generation without degrading text quality.

## 2. Related Work

### 2.1. Token-level Decoding Controls and Degeneration

The notion of *degeneration* in neural text generation was popularized by Holtzman et al. (2020), which highlighted that the most probable token is not always the best continuation. A wide range of decoding heuristics seek to improve generation by reshaping the next-token distribution (e.g., truncation, penalties, and sampling variants). Locally typical sampling (Meister et al., 2023) selects tokens whose surprisal is close to the local entropy, aiming to avoid both overly likely and overly unlikely choices. These methods operate at the token/probability level; our focus is complementary: we study the internal dynamics underlying mode collapse and propose a state-space intervention. The gap between token-level, microscopic decoding and long-range, macroscopic generation behavior is shared by many other fields, such as LLM hallucination studies (Huang et al., 2025).

### 2.2. Repetition and Mode Collapse in Text Generation

Prior work has proposed diverse explanations for repetition and looping in autoregressive decoding. Some accounts attribute repetition to structural properties of token-transition probabilities (high-inflow patterns) (Fu et al., 2021) or to self-reinforcing feedback in next-token prediction (Xu et al., 2022). Others emphasize data-driven origins of repetition in training corpora (Li et al., 2023), or propose pattern-detection and distribution-adjustment schemes to avoid structural repetitions in constrained domains such as code (Dong et al., 2025). Our work provides a different perspective: mode collapse is associated with geometric collapse of the internal trajectory, related to the dynamical-system aspects of the LLM.

### 2.3. Geometric Diagnostics of Internal Dynamics

Recent work has explored geometric characterizations of language and LLM representations, including fractal- (Doxas et al., 2010; Alabdulmohsin et al., 2024; Du & Tanaka-Ishii, 2025) and intrinsic-dimension perspectives (Campadelli et al., 2015; Aghajanyan et al., 2021). More broadly, long-range statistical diagnostics such as scaling laws and repeated-subsequence entropy have been used to evaluate whether generated text preserves natural-language structure beyond local fluency (Takahashi & Tanaka-Ishii, 2019; Tanaka-Ishii, 2026). In parallel, mechanistic interpretability research has developed tools for reading out intermediate model beliefs, such as the tuned lens (Belrose et al., 2023), which decodes layerwise latent predictions. These approaches share the goal of connecting observable behavior to internal structure; our contribution is to provide a dynamical-systems perspective and use correlation dimension as a trajectory-level diagnostic of state-space accessibility and relate it directly to mode collapse in decoding.

### 2.4. Inference-time Interventions

RMR is most closely related to inference-time activation interventions (Zou et al., 2023; Turner et al., 2023; Meng et al., 2023), but differs in goal and mechanism: we target persistent, self-reinforcing directions identified by a bounded-spectrum generalized eigenvalue problem, and regulate the value cache to reduce mode collapse. In addition, we study the long-range behavior of LLM from a dynamical-systems perspective, rather than for a specific task.

## 3. The Core Idea

We model autoregressive generation as an evolution of an internal state, and our core hypothesis is that mode collapse is a symptom of an underlying *geometric collapse*: during decoding, the internal trajectory becomes confined to a low-dimensional region of representation space, i.e., state-space accessibility collapses. A canonical and easily observable manifestation is *explicit looping*, where the model enters a periodic regime and repeatedly produces the same token or short token sequence. See examples in Appendix C. This view shifts the focus from token-level probabilities to the geometry of the underlying dynamics, and motivates intervening directly in the state evolution.

We use a minimal dynamical model to illustrate how such accessibility collapse can arise and how a simple regulation principle can prevent it. We then verify in real LLM decoding that geometric collapse (measured by correlation dimension) is closely associated with symbolic mode collapse (Section 4), and finally introduce an LLM-specific method (Section 5).

### 3.1. A Minimal Model

We present a minimal dynamical model to illustrate the geometric origin of LLM looping. The model is based on

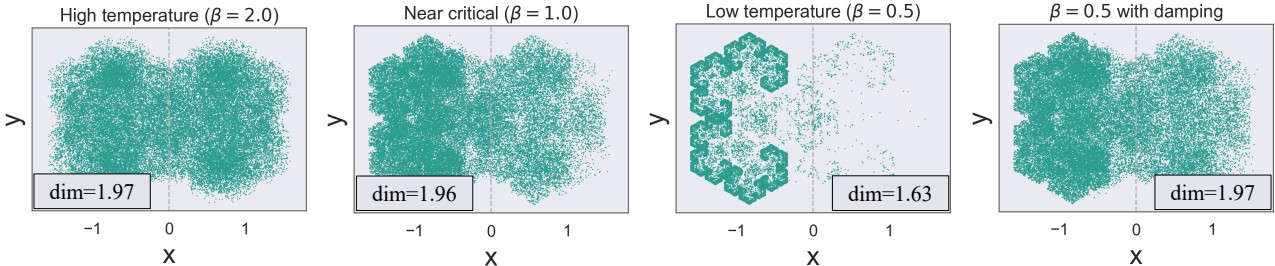

*Figure 1.* Trajectories of the state-dependent IFS in Eq. (13). We use $r_i = 0.6$ $(\forall i)$, $\mathbf{b}_i = [2g_i, 0]^\top$, and a rotation $\mathbf{O}_i$ with angle $\theta_i = -g_i\pi/3$. (a–c) Different inverse temperatures $\beta$: (a) $\beta = 2.0$, (b) $\beta = 1.0$, (c) $\beta = 0.5$. (d) Regulation implemented as weak damping (Eq. (9)) applied to the history-dependent variable $m_t$.

an *iterated function system* (IFS) (HUTCHINSON, 1981; Barnsley & Demko, 1985; Feng & Hu, 2009), augmented with state-dependent map selection to capture temperature-induced phase transitions. We refer to this construction as a *state-dependent IFS*. Figure 1(a–c) visualizes the system's long-term behavior under different temperatures.

For clarity, we consider a two-dimensional state space $X = \mathbb{R}^2$. The system consists of $2m$ contraction maps $\{f_i : \mathbb{R}^2 \to \mathbb{R}^2\}_{i=1}^{2m}$. The first $m$ maps contract the trajectory toward the left half-plane, while the remaining $m$ maps contract it toward the right. Let $g_i \in \{-1, +1\}$ denote the group that map $i$ belongs to, setting $g_1 = \ldots = g_m = -1$, $g_{m+1} = \ldots = g_{2m} = 1$.

Let $\mathbf{x}_t = (x_t, y_t)$ denote the system state at time $t$, and let $\pi_t$ be a time-dependent distribution over the maps. The system evolves as

$$\mathbf{x}_{t+1} = f_i(\mathbf{x}_t) = r_i\mathbf{O}_i\mathbf{x}_t + \mathbf{b}_i, \qquad i \sim \pi_t, \tag{1}$$

$$\pi_t(i) = \exp\left(\frac{m_t g_i}{\beta}\right) \Big/ \sum_{j=1}^{2m} \exp\left(\frac{m_t g_j}{\beta}\right), \quad \forall i,$$

where $0 < r_i < 1$ is the contraction factor, $\mathbf{O}_i$ is a rotation matrix, and $\mathbf{b}_i$ is a translation vector. The quantity $m_t$ denotes a history-dependent feedback term defined as

$$m_t = \frac{1}{t}\sum_{s=1}^{t} x_s, \tag{2}$$

representing the cumulative bias of the trajectory along the horizontal axis. $g_i$ decides the direction of the bias that points to either group of maps.

When the historical average $m_t$ becomes negative, probability mass in $\pi_t$ shifts toward maps contracting into the left half-plane, which further reinforces the trajectory bias. This positive feedback mechanism is analogous to the mean-field Ising model for explaining the phase transition in magnetic materials. There exists a critical temperature $\beta_0 > 0$ such that when $\beta > \beta_0$, the system admits a unique ergodic invariant measure (i.e., a single connected component), whereas

for $\beta < \beta_0$, two stable long-run regimes emerge, corresponding to trajectories confined to the left or right half-plane.

Figures 1(a-c) illustrate this behavior. At high temperature (a-b), the trajectory explores the full state space and sufficiently fills the ambient 2D space. Below the critical temperature (c), the trajectory becomes trapped in a local attractor associated with a sub-IFS. This transition provides a minimal geometric explanation for accessibility collapse: the accessible region shrinks and the fractal dimension, introduced in Section 4.1, becomes lower in (c) than in (a-b). The resulting trajectory is more regular and predictable, analogous to mode collapse behaviors in LLM generation, such as repetition and looping. Although LLMs do not strictly satisfy contraction conditions, this model captures the essential near-looping regime where effective trajectories contract toward local, low-dimensional "attractors."

This model is not intended as a faithful derivation of Transformer KV-cache dynamics. Rather, it is a deliberately restricted system that isolates one mechanism shared with autoregressive decoding: the state produced by previous steps influences future transition probabilities, allowing small historical biases to become self-reinforcing over long horizons. We therefore use it only as a qualitative model of accessibility collapse and critical transition; the LLM-specific mechanism is studied empirically in Sections 4 and 5.

### 3.2. Restoring State-Space Accessibility via Regulation

In the minimal model, accessibility collapse arises from the accumulation of the history-dependent bias term $m_t$, which becomes dynamically dominant only in the low-temperature regime. Suppressing this accumulation can therefore prevent geometric collapse even when the temperature is low.

Concretely, we introduce a weak regulation factor $\eta > 0$ (which can be viewed as damping),

$$m_t \leftarrow (1 - \eta)\, m_t, \tag{3}$$

at every timestep $t$, which limits long-term bias accumu-

lation while preserving short-term dynamics. Figure 1(d) shows that even a very small regulation strength ($\eta = 10^{-4}$) suffices to restore state-space accessibility at low temperature.

From a dynamical perspective, variables such as $m_t$ represent directions of persistent temporal influence. Near critical regimes, these directions evolve significantly more slowly than the fast-mixing components of the system, and therefore dominate long-term behavior. Such emergent persistence has been studied across multiple theoretical frameworks, including nonequilibrium statistical mechanics (Mori, 1965; Zwanzig, 1960; Te Vrugt & Wittkowski, 2019; Chorin et al., 2000; Hohenberg & Halperin, 1977), dynamical-systems and multiscale analysis (Carr, 2012; Fenichel, 1979; Kuehn, 2011), and synergetics/self-organization (Haken, 1977). Importantly, slow behavior here is not assumed a priori, but emerges as a consequence of geometric collapse.

While this form of regulation is straightforward in low-dimensional systems, identifying persistent directions from high-dimension observations remains challenging. In the following section, we introduce a practical method to detect and regulate such geometric collapse during LLM decoding.

## 4. Geometric Collapse in LLM Decoding

In this section, we move from the abstract core idea to empirical evidence in real LLM decoding. We start from a canonical symbolic manifestation of mode collapse—explicit looping—and then show how the correlation dimension captures an underlying geometric collapse of the internal trajectory that accompanies such failures.

Figure 2 highlights why a geometric probe is useful: even when token-level proxies change gradually or are sensitive to the particular realization, the correlation dimension provides a direct and robust signal of reduced accessibility. In intermediate regimes, generated text often already shows degeneration—template reuse, semantic stagnation, or local variations without new content—before entropy or Distinct-$n$ registers a sharp change.

In many nonlinear systems, long-term behavior is confined to a restricted subset of the state space. Classical dynamical systems theory describes such phenomena using attractors; however, for high-dimensional stochastic systems such as large language models, attractors are generally neither well-defined nor directly observable. Instead, what can be empirically assessed is the *effective accessibility* of the state space over finite time horizons.

While the state space $X$ can be high-dimensional, a large region of the space is often statistically inaccessible, i.e., with zero probability that a trajectory will visit the region

sufficiently often. Therefore, the notion of accessibility can be formalized via a probability distribution (measure) $\mu$ over $X$. Moreover, $\mu$ can be *invariant* under $f$, and $\mu$ is called an *invariant measure* of the system.

From this perspective, accessibility reflects the number of (dynamically) active degrees of freedom that remain active during generation. Normal text generation corresponds to trajectories that continue to explore a large, high-dimensional region, whereas mode collapse corresponds to a progressive trapping into a much lower-dimensional subset. A simple illustrative example is provided by the system $(r, \theta)$ in polar coordinates: $\dot{r} = 1 - r$, $\dot{\theta} = \text{constant} \neq 0$, whose trajectory converges to the unit circle. While the ambient space is two-dimensional, the long-term motion is effectively one-dimensional.

In this work, we characterize state-space accessibility using the *correlation dimension*, a trajectory-wise fractal dimension originally introduced in nonlinear time-series analysis (Grassberger & Procaccia, 1983; Osborne & Provenzale, 1989) and applied to many real-world systems (Small & Judd, 1998; Lacasa & Gómez-Gardenes, 2013; Du & Tanaka-Ishii, 2024). Correlation dimension quantifies how the number of accessible states scales with resolution, and thus provides a direct geometric measure of effective dimensionality. In Section 4.1, we formally define this quantity and show that mode collapse in large language models is closely associated with a pronounced low-dimensional collapse of the generation trajectory.

### 4.1. Fractal Dimension

Fractal dimension generalizes the notion of dimension to non-integer values. Non-integer dimension typically appears in fractals, which are geometric structures that exhibit self-similarity at different scales.

The correlation dimension of a trajectory $\{x_1, x_2, \ldots\}$ is defined as the scaling exponent $d$ in the relation

$$C_t(\varepsilon) \propto \varepsilon^d \quad \text{as } \varepsilon \to 0, t \to \infty \qquad (4)$$

where

$$C_t(\varepsilon) = \frac{2}{t(t-1)} \sum_{i=1}^{t-1} \sum_{j=i+1}^{t} \mathbf{1}\left(\|x_i - x_j\| < \varepsilon\right) \quad (5)$$

is called the correlation sum, and represents the fraction of point pairs whose distance is smaller than $\varepsilon$. In practice, the dimension is estimated from the slope of $\log C_t(\varepsilon)$ versus $\log \varepsilon$ over a finite scale range; a more rigorous discussion is provided in Appendix B.

To analyze the evolution of generation dynamics, we compute correlation dimension online at each timestep. Specifically, we define the *finite-time correlation dimension* as follows.

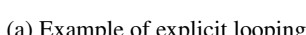

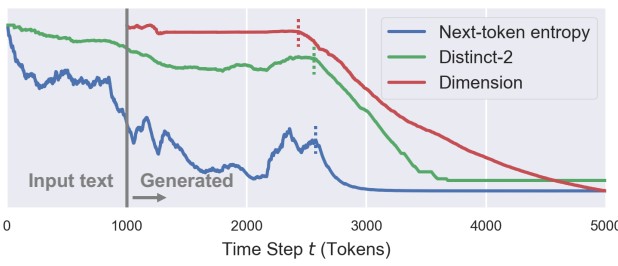

(a) Example of explicit looping

(b) Correlation dimension vs. token-level proxies

*Figure 2.* From explicit looping to geometric collapse. (a) An example of explicit mode collapse (looping) with Qwen3-4B-Base at temperature 0.5, chosen to make the visible token-level loop appear early and clearly. (b) A separate trajectory generated at temperature 1.0. Generated part starts at $t = 1000$. Next-token entropy (blue) and Distinct-2 (green) provide token-level proxies of diversity, whereas correlation dimension (red) directly measures state-space accessibility; it drops sharply as the trajectory concentrates into a low-dimensional regime.

**Definition 4.1** (Finite-Time Correlation Dimension). Given a trajectory $x_1, \ldots, x_t$, the finite-time correlation dimension $d_t^{(\varepsilon_0, \varepsilon_1)}$ is defined as the best-fit slope of $\log C_t(\varepsilon)$ versus $\log \varepsilon$ for $\varepsilon$ sampled log-uniformly from $(\varepsilon_0, \varepsilon_1)$.

Although the naive computation of $C_t$ requires $O(t^2)$ time, it admits an efficient online update:

$$C_{t+1}(\varepsilon) = \frac{t-1}{t+1} C_t(\varepsilon) + \frac{2}{t(t+1)} \sum_{i=1}^{t} \mathbf{1}\left(\|x_i - x_{t+1}\| < \varepsilon\right),$$

which allows $d_t$ to be computed in $O(t)$ time.

For LLMs, we used the next-token log-probability vector as dynamical-system state $x_t$, and measured the correlation dimension using the log-probability vector sequence. Detailed settings are provided in Appendix B.1.

### 4.2. Geometric Collapse in LLM Generation

Explicit looping (Figure 2(a)) is a representative manifestation of geometric collapse in LLM, corresponding to extremely limited state-space accessibility and thus exceptionally low correlation dimension.

Previous studies often assess looping with symbolic quantities like next-token conditional entropy and Distinct-$n$ (i.e., proportion of distinct $n$-grams in the generation sequence), but both are sensitive to intrinsic linguistic variation.

In Figure 2(b), entropy fluctuates well before looping and Distinct-2 drops only after repetition becomes explicit. In contrast, correlation dimension remains stable during normal generation and decreases only when the trajectory exhibits geometric collapse.

Many studies have observed that lowering generation temperature increases the risk of looping (Nakaishi et al., 2024; Pipis et al., 2025), and Figures 3(a-b) verify this observation and show a relationship between looping rate and the temperature. Similar results are observed in Figure 3(c-d)

where the temperature is adaptively adjusted while maintaining a constant next-token entropy. In (a) and (c), mode collapse is measured by the exact looping rate, while in (b) and (d), we report the mean correlation dimension over multiple generation runs. A monotonic decrease in dimension is observed along generation, with significantly stronger collapse at low temperatures or low entropy constraints. A clear correspondence between looping rate and correlation dimension is observed in all decoding conditions.

**Limitations of symbolic explanations.** The collapse at low entropy constraints shows limitations of the symbolic explanations of mode collapse (see Section 2), that the generation process enters a process where entropy is gradually reduced to 0. In the entropy-locked decoding experiment, however, the collapse still occurs if the entropy constraint goes below a crticial value.

On the other hand, even if the generation entropy is set to a relatively high constant value (e.g., 2.0) when exact looping is less observed (Figure 3(c)), geometric collapse still occurs (as in (d)), manifesting in more implicit forms, such as template repetition or semantic looping. This phenomenon corresponds to what has been termed *degeneration* in Holtzman et al. (2020), which includes more subtle failures such as incoherence or blandness. Representative examples of this softer form of collapse are provided in Appendix C.

This indicates the unique value of correlation dimension in reflecting mode collapse in LLM generation. Different from symbolic quantities like entropy or Distinct-$n$, correlation dimension captures the active degrees of freedom in a generation trajectory.

## 5. Reinforced Mode Regulation for LLM

As demonstrated in Sections 3, geometric collapse is a key mechanism underlying mode collapse, and regulating a small number of persistent components can effectively

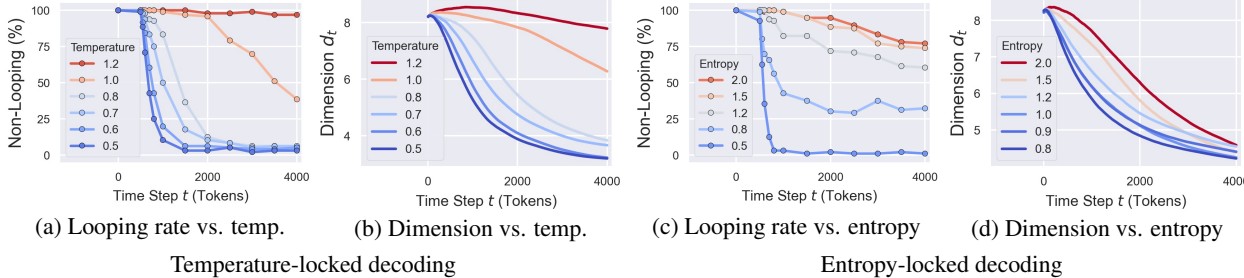

(a) Looping rate vs. temp.   (b) Dimension vs. temp.

Temperature-locked decoding

(c) Looping rate vs. entropy   (d) Dimension vs. entropy

Entropy-locked decoding

*Figure 3.* Correlation dimension tracks mode collapse across decoding conditions. As randomness is reduced (lower temperature or lower entropy target), explicit looping becomes more frequent, and correlation dimension decreases in tandem, indicating progressive concentration into low-dimensional regimes.

prevent such collapse.

In this section, we propose *Reinforced Mode Regulation* (RMR), a lightweight state-space intervention for Transformer-based LLMs. RMR regulates a small set of directions in the value-cache space that exhibit unusually strong temporal persistence, thereby preventing the cache trajectory from concentrating along a single persistent direction. Concretely, RMR can be implemented as a low-rank damping operator applied to the value cache along these directions. These directions are identified by a generalized eigenvalue problem whose eigenvalues admit a natural, bounded scale; the eigenvalue directly quantifies persistence and enables principled thresholding (damping).

**Key-Value Cache as State of Transformer** In a Transformer, the generation state at timestep $t$ is fully represented by the key-value (KV) cache across layers,

$$\{(\mathbf{K}_t^{(l)}, \mathbf{V}_t^{(l)})\}_{l=1}^{L}, \qquad \mathbf{K}_t^{(l)}, \mathbf{V}_t^{(l)} \in \mathbb{R}^{t \times D}, \qquad (6)$$

where batch and head dimensions are omitted for clarity.

At each decoding step, the cache is extended as

$$\mathbf{K}_t = [\mathbf{K}_{t-1}; \mathbf{k}_t], \qquad \mathbf{V}_t = [\mathbf{V}_{t-1}; \mathbf{v}_t], \qquad (7)$$

and the value cache is aggregated through attention:

$$\text{Attention}(\mathbf{q}_t; \mathbf{K}_t, \mathbf{V}_t) = \text{softmax}\left(\frac{\mathbf{q}_t \mathbf{K}_t^\top}{\sqrt{D}}\right) \mathbf{V}_t. \qquad (8)$$

We focus on the value-cache matrices $\mathbf{V}_t$. The discussion below considers an arbitrary layer and applies analogously to other layers. We do *not* regulate the key-cache matrices, since modifying keys changes attention weights and can disrupt the global temporal dependency structure.

We view $\mathbf{V}_t \in \mathbb{R}^{t \times D}$ as a sequence of vectors collected along the temporal dimension. When generation approaches a degenerative regime, we assume that geometric collapse concentrates along a low-dimensional subspace, represented

by an orthonormal basis $\mathbf{U} \in \mathbb{R}^{D \times c}$. Damping is then applied by suppressing the projection onto this subspace:

$$\mathbf{V}_t \leftarrow (\mathbf{I} - \eta \mathbf{\Gamma}) \, \mathbf{V}_t (\mathbf{I} - \mathbf{U}\mathbf{U}^\top), \qquad (9)$$

where $\eta > 0$ controls the damping strength and $\mathbf{\Gamma} = \text{diag}(\gamma^t, \gamma^{t-1}, \ldots, \gamma^1)$ applies exponentially weaker damping to older timesteps, with $\gamma$ set to 0.995 in our experiments.

To identify the dominant persistent directions, we estimate $\mathbf{U}$ by solving the generalized eigenvalue problem

$$\mathbf{\Sigma}_\Delta \mathbf{u} = \lambda \, \mathbf{\Sigma} \mathbf{u}, \qquad (10)$$

where

$$\mathbf{\Sigma} = \mathbb{E}_t \big[ (\mathbf{v}_t - \bar{\mathbf{v}})(\mathbf{v}_t - \bar{\mathbf{v}})^\top \big], \qquad (11)$$

$$\mathbf{\Sigma}_\Delta = \mathbb{E}_t \Big[ \text{sym}\big( (\mathbf{v}_{t+1} - \bar{\mathbf{v}})(\mathbf{v}_t - \bar{\mathbf{v}})^\top \big) \Big], \qquad (12)$$

where $\text{sym}(\mathbf{A}) := \frac{1}{2}(\mathbf{A} + \mathbf{A}^\top)$, with $\mathbf{v}_t \in \mathbb{R}^D$ denoting the $t$-th row of $\mathbf{V}_t$ and $\bar{\mathbf{v}}$ its temporal mean.

Under stationary conditions, the spectrum is bounded with $|\lambda| \leq 1$. Large eigenvalues ($\lambda \approx 1$) correspond to directions with slow temporal decay, indicating strong persistence. We select the top-$c$ eigenvectors to form $\mathbf{U}$. The required statistics are estimated online using exponentially weighted moving averages (decay rate $\gamma = 0.99$), and the top eigenvectors are efficiently computed via power iteration (Appendix D). The eigenvalues can be understood as the *auto-correlation* coefficient of $\mathbf{v}_t$ along different directions. Therefore, our method can be naturally generalized to other LLM architectures, such as state-space models (Gu & Dao, 2024).

**Eigenvalue Threshold and Regulation** Thresholding is crucial for stable intervention: because Eq. (10) induces a bounded spectrum ($|\lambda| \leq 1$ under stationarity), any threshold has a consistent meaning across models, layers, and timesteps. We set an eigenvalue threshold $\lambda_{\min} = 0.8$ and

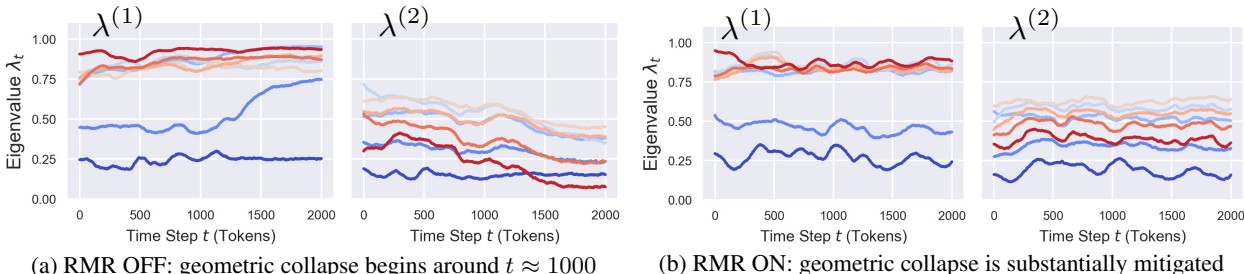

(a) RMR OFF: geometric collapse begins around $t \approx 1000$      (b) RMR ON: geometric collapse is substantially mitigated

*Figure 4.* Evolution of the top-2 generalized eigenvalues in the value cache across Transformer layers (color-coded from blue to red), reporting the mean over attention heads. (a) Without RMR, geometric collapse begins around $t \approx 1000$: the leading eigenvalue rises toward 1 while the second eigenvalue quickly drops toward 0, indicating a widening spectral gap and dominance by a single persistent mode. (b) With RMR, the leading eigenvalue in each layer is regulated below the threshold ($\lambda_{\min} = 0.8$), maintaining a stable spectrum and preventing collapse.

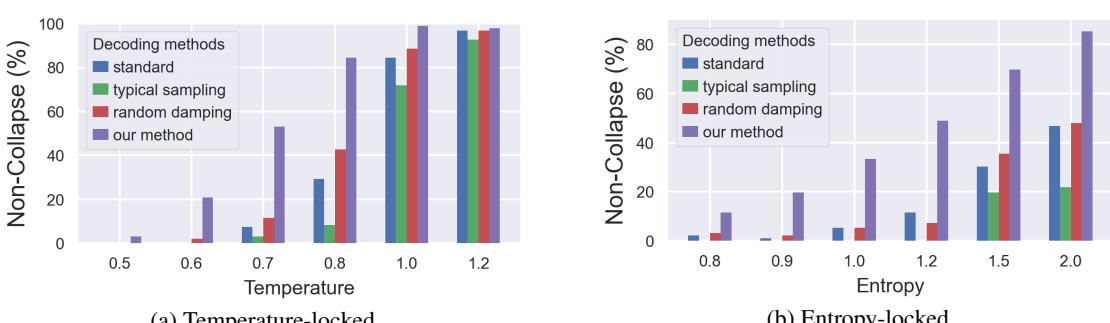

(a) Temperature-locked          (b) Entropy-locked

*Figure 5.* Non-collapse rates under controlled randomness. (a) Temperature-locked decoding and (b) entropy-locked decoding, comparing standard decoding, typical sampling, random regulation, and RMR. A completion is counted as non-collapse if its correlation dimension remains above 8 over 1,000 generated tokens of HEIDEGGER.

define $c$ as the number of eigenvalues with $\lambda > \lambda_{\min}$. In practice, most eigenvalues are small, so we estimate only the top-8 eigenvalues/eigenvectors and regulate those exceeding $\lambda_{\min}$. For the value-cache component along the selected subspace, we use a fixed regulation strength $\eta = 0.7$ (implemented as damping in Eq. (9)). Compared to fixing $c$, thresholding is more stable: even if regulation is applied at consecutive steps, "fast" directions with small $\lambda$ remain unaffected, allowing mid-level persistent variables (e.g., topic and style) to be preserved.

In practice, we attenuate only directions whose eigenvalues exceed the threshold and reduce them to slightly below $\lambda_{\min}$. This eigenvalue threshold has a concrete meaning (due to the bounded spectrum) and is therefore more reliable than regulating a fixed number $c$ of directions: it avoids unnecessarily suppressing moderately persistent components and reduces unintended side effects. Empirically, performance is relatively insensitive to the regulation interval; applying damping once every 10 decoding steps already works well in our experiments.

**Spectral Structure and Effects of Regulation** Figure 4 shows the evolution of the top-2 generalized eigenvalues

(Eq. (10)) across Transformer layers, averaged over all attention heads (see Appendix E for additional details, including both mean and maximum over heads). In Figure 4(a) (RMR off), the largest eigenvalue stays close to 1 in most layers (except near the input embedding). Around the onset of mode collapse ($t \approx 1000$), the top-1 eigenvalue in some layers increases while the top-2 eigenvalue quickly decreases toward 0, indicating that the dynamics become increasingly dominated by a single persistent direction (a growing spectral gap).

Figure 4(b) shows a run with RMR where collapse is prevented. Regulation keeps the top-1 eigenvalue in each layer below the threshold $\lambda_{\min}$, and both top-1 and top-2 eigenvalues fluctuate around stable values throughout generation.

## 6. Experimental Setup

Throughout the paper we use standard top-$k$ and top-$p$ filtering to improve generation quality, with $k = 50$ and $p = 0.9$ (i.e., restricting next-token candidates to at most 50 tokens with cumulative probability at least 0.9). In entropy-locked experiments, the target entropy is computed over the same top-$k$ candidate set, so the reported entropy reflects uncer-

tainty after the top-$k$ restriction.

Unless otherwise noted, results in the main text are obtained from a fixed prompt: the first 1,000-token segment of HEIDEGGER from the SEP dataset, using Qwen3-4B-Base (Yang et al., 2025). Comprehensive results across additional prompts and larger, instruction-tuned models are provided in Appendix F.1 and Appendix F.2.

In addition to RMR, we also consider *typical sampling* (Meister et al., 2023) as a baseline. Typical sampling favors tokens whose surprisal is close to the distribution's "typical" value (near the local entropy), effectively filtering out both overly likely and overly unlikely tokens before sampling. Following Meister et al. (2023), we set the typical cumulative probability to 0.2.

## 7. Results

**Mitigating Mode Collapse via Geometric Regulation**
Figure 5 compares RMR with standard decoding across different levels of randomness. Panel (a) fixes the generation temperature and panel (b) fixes the next-token entropy to the constants shown on the horizontal axis. Non-collapse rate is computed as the fraction of generated 1,000-token completions of HEIDEGGER for which the correlation dimension is above 8. We use this threshold as a geometric diagnostic rather than as a complete definition of text quality; the same experimental section also reports token-level and judge-based evaluations that do not use correlation dimension.

Across all temperature-locked and entropy-locked settings, RMR consistently achieves substantially higher non-collapse rates than standard decoding. The gain is most pronounced near the critical regime where baseline decoding starts to fail (temperature $\approx$ 0.7–0.8). For example, at temperature 0.7, the non-collapse rate increases from 8% (standard) to 56% with RMR. In the entropy-locked setting, RMR is the only approach that maintains a sizable non-collapse rate under extremely low entropy rates (e.g., entropy $=$ 1.0), where standard decoding nearly always collapses (about 5% non-collapse versus 33% for RMR at entropy $=$ 1.0).

**Baselines: Typical Sampling and Random Regulation**
Figure 5 also includes two baselines. Typical sampling (Meister et al., 2023) does not prevent collapse in the low-randomness regime and can underperform standard decoding, consistent with its tendency to reduce effective sampling uncertainty and push generation toward low-entropy collapse. Random regulation—regulating a randomly chosen value-cache subspace with the same rank and strength—also falls short. These results support that RMR's benefit comes from targeting the specific persistent subspace rather than

generic interventions.

**External Collapse Indicators**    Although our primary non-collapse rate is based on correlation dimension, we also report collapse indicators that do not use the proposed geometric diagnostic. In Figure 3(a,c), exact looping is measured directly from the generated token sequence and follows the same low-randomness transition as the dimension drop. Figure 2 contrasts correlation dimension with next-token entropy and Distinct-2: these token-level quantities are less temporally precise, but they provide independent evidence of the surface degeneration associated with collapse. The examples in Appendix C further show that collapse can appear as template repetition or semantic stagnation even before it becomes an exact token loop.

**Text Quality Evaluation**    An important concern is whether intervening in the KV cache degrades the surface quality of text when generation is *not* collapsing. To isolate potential side effects, we evaluate only non-collapse completions (correlation dimension $>$ 8) generated at relatively high temperatures (1.0 or 1.2), where standard decoding is typically stable. Table 1 reports LLM-as-a-Judge scores comparing standard decoding and RMR. Across models and criteria, the score distributions are statistically indistinguishable (Kolmogorov-Smirnov (K-S) $p$-values well above 0.05), suggesting that regulating the eigenvalue-thresholded subspace does not introduce noticeable changes to surface properties such as coherence, syntax, and information progression. The full rating prompt is provided in Appendix G.

## 8. Limitations

Non-autoregressive LLMs (e.g., diffusion-based models) are not considered in this work. RMR also assumes access to internal Transformer states (the KV cache) during decoding, which may be infeasible in black-box settings and requires integration into inference stacks. While RMR is lightweight, it adds overhead for maintaining running statistics and estimating dominant eigenvectors, which can matter under strict latency constraints or for very large models.

We use a fixed eigenvalue threshold and regulation strength throughout our experiments and find the method to be stable under this simple configuration; in particular, the threshold $\tau = 0.8$ is effective across all tested settings. The main-text experiments use a fixed long-form continuation setting for clarity; the appendix broadens this to additional SEP prompts and instruction-tuned models, while broader coverage of multi-turn dialogue, RAG, code generation, and reasoning tasks remains future work. Finally, our evaluation focuses on mode collapse as characterized by correlation dimension and related repetition indicators; interactions with other failure modes, safety behaviors, and downstream task

*Table 1.* LLM-as-a-Judge scores for non-collapse texts. Completions are generated in a temperature-locked setting with (a) 1,000 tokens and (b) 4,000 tokens; only non-collapse samples are included. Large $p$-values indicate statistically indistinguishable score distributions (larger is better).

**(a) Generations of 1,000 tokens**

| Method
(Non-collapse rate) | standard
(84%) | RMR
(99%) | p-value |
|---|---|---|---|
| **Deepseek-R1-14B** | | | |
| Coherence | 8.26 (1.2) | 8.27 (1.0) | 0.33 |
| Syntax | 6.98 (1.1) | 7.07 (0.9) | 0.97 |
| Progress | 7.21 (1.2) | 7.14 (1.1) | 0.22 |
| **GPT-5-mini** | | | |
| Coherence | 6.32 (1.9) | 6.25 (2.1) | 0.55 |
| Syntax | 6.10 (1.6) | 6.07 (1.9) | 0.98 |
| Progress | 2.51 (1.8) | 2.22 (1.8) | 0.72 |

**(b) Generations of 4,000 tokens.** $p$-values are computed between standard decoding (temp. $= 1.2$) and RMR (temp. $= 1.0$).

| Method | standard | | RMR | p-value |
|---|---|---|---|---|
| Temperature
(Non-collapse rate) | 1.0
(6%) | 1.2
(55%) | 1.0
(99%) | |
| **Deepseek-R1-14B** | | | | |
| Coherence | 5.75 | 7.66 | 7.34 | 0.85 |
| Syntax | 5.50 | 6.67 | 6.70 | 0.99 |
| Progress | 5.03 | 6.69 | 6.36 | 0.28 |
| **GPT-5-mini** | | | | |
| Coherence | 5.08 | 6.40 | 6.23 | 0.14 |
| Syntax | 4.33 | 5.68 | 5.95 | 0.58 |
| Progress | 2.00 | 2.73 | 2.78 | 0.99 |

performance remain for future work.

## 9. Conclusion

We revisited mode collapse in autoregressive text generation from a dynamical-systems perspective and linked symbolic failures to *geometric collapse* of the internal trajectory (reduced state-space accessibility). Guided by this view, we proposed *Reinforced Mode Regulation* (RMR), which identifies and attenuates the dominant persistent subspace in the Transformer value cache via a bounded-spectrum generalized eigenvalue formulation and thresholding. Across multiple LLMs, RMR substantially improves non-collapse rates across temperatures and entropy targets, and preserves surface text quality in non-collapse regimes.

RMR is designed and evaluated as a decoding-time intervention. More broadly, trajectory-level diagnostics such as correlation dimension may also be useful for analyzing training outcomes, which remains a future work.

## Acknowledgements

This work is supported by JST CREST, Japan, Grant Number JPMJCR2114.

## Impact Statement

This paper presents work whose goal is to advance the field of Machine Learning. There are many potential societal consequences of our work, none which we feel must be specifically highlighted here.

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

## A. Iterated Function Systems

An iterated function system (IFS) consists of a collection of contraction maps $\{f_i : X \to X\}_{i=1}^m$ on a metric space $X$ and a probability distribution $\pi$ over the maps (with $\pi_i > 0$). At each discrete time step, we sample an index $i \sim \pi$ and update $\mathbf{x}_{t+1} = f_i(\mathbf{x}_t)$. The simplest example is the *self-similar* (affine) IFS:

$$\mathbf{x}_{t+1} = f_i(\mathbf{x}_t) = r\mathbf{x}_t + \mathbf{b}_i, \qquad i \sim \pi, \tag{13}$$

where $r$ ($0 < r < 1$) is the contraction factor and $\mathbf{b}_i \in \mathbb{R}^n$ is a translation. Under standard assumptions, such affine IFS admit a unique attractor and an ergodic invariant measure $\mu$ supported on it. Under mild conditions, the fractal dimension of $\mu$ satisfies

$$d(\mu) = \min\left\{\frac{h_\pi}{-\log r}, \ \dim(X)\right\}, \tag{14}$$

where $h_\pi$ is the Shannon entropy (Feng & Hu, 2009) of $\pi$.

## B. Fractal Dimension and Correlation Dimension

Fractal dimension can be defined with respect to a set (e.g., Hausdorff dimension, box-counting dimension) or a measure (e.g., information dimension, correlation dimension). The distinction between the two is whether the distribution of the points in the set is considered. Measure-based fractal dimension is typically used in the context of dynamical systems, where how *frequently* a trajectory visits a region is investigated. In many cases, the two classes of fractal dimensions are close in value.

Below is a definition of measure-based fractal dimension.

**Definition B.1** (Fractal Dimension of Measure). For a probability measure $\mu$ on a metric space $X$, the local fractal dimension of $\mu$ is defined as

$$d(\mu, \mathbf{x}) = \lim_{\varepsilon \to 0} \frac{\log \mu\big(B(\mathbf{x}, \varepsilon)\big)}{\log \varepsilon}, \tag{15}$$

where $B(\mathbf{x}, \varepsilon)$ is the closed ball centered at $\mathbf{x}$ with radius $\varepsilon$. If $d(\mu, \mathbf{x})$ is constant for $\mu$-almost all $\mathbf{x} \in X$, we denote the value by $d(\mu)$ and call it the *fractal dimension* of $\mu$.

**The Grassberger–Procaccia estimator and Correlation Dimension**    In practice, $\mu$ is usually intractable and manifest only as the limit of the empirical measure. Correlation dimension is essentially an estimator of the measure of the ball $B(\mathbf{x}, \varepsilon)$ in Eq. (15).

Let $t$ denote the trajectory length. The Grassberger–Procaccia (GP) method (Grassberger & Procaccia, 1983) estimates the correlation sum

$$C_t(\varepsilon) = \frac{2}{t(t-1)} \sum_{1 \le i < j \le t} \mathbb{I}(\|\mathbf{x}_i - \mathbf{x}_j\|_2 \le \varepsilon), \tag{16}$$

which approximates $\mu(B(\mathbf{x}, \varepsilon))$ up to constants for small $\varepsilon$ and large $t$. The (correlation) dimension is then estimated from the slope of $\log C_t(\varepsilon)$ versus $\log \varepsilon$ over an appropriate scaling range. This estimator is proposed by Grassberger and Procaccia (Grassberger & Procaccia, 1983), and called the GP estimator by convention. See (Pesin, 1993) for a rigorous mathematical definition of correlation dimension.

### B.1. Estimation Details of Correlation Dimension

We estimate the correlation dimension of a generation trajectory $\mathbf{x}_1, \ldots, \mathbf{x}_t$ by using the next-token log-probability vector sequence as the state, following (Du & Tanaka-Ishii, 2025). Specifically, at each time step $t$ we define

$$\mathbf{x}_t = \log \mathbb{P}(w_t | w_{t-1}, \ldots, w_1), \quad w_t \in \mathcal{V}, \tag{17}$$

where $w_t$ is the $t$-th token in the generation sequence, $\mathcal{V}$ is the vocabulary.

To reduce the computational cost due to the large vocabulary size (typically above 100,000), we use a projection-based approximation inspired by Marstrand's projection theorem (Marstrand, 1954; Falconer, 2004): we first group the dimensions

in $\mathcal{V}$ into $K = 10{,}000$ bins using $\Phi(i) = i \bmod K$, and add up the log-probabilities in $\mathbf{x}_t$ within each bin to get a 10,000-dimensional vector, and use it in replacement of $\mathbf{x}_t$.

The finite-time correlation dimension defined in Definition 4.1 specifies a scaling range $(\varepsilon_0, \varepsilon_1)$, in which the correlation sum $C_t(\varepsilon)$ is expected to scale quasi-linearly with $\varepsilon$. For Qwen3-4B-Base, we find the distances between timesteps to range from about 100 to 10000. Therefore, we set $\varepsilon_0 = 500$ and $\varepsilon_1 = 1200$, which is in the middle of the range in the log scale. This setting differs from (Du & Tanaka-Ishii, 2025), in which a smaller range at around $(100, 500)$ is used. We found the setting in (Du & Tanaka-Ishii, 2025) to be very sensitive to incremental changes in the generation trajectory, unsuitable for temporal monitoring of correlation dimension.

# C. Additional Examples

···Dasein is not a detached observer of the world, but is always engaged in practical activities and projects. Dasein is always already involved in a network of practical concerns and relationships. This is what Heidegger means by Dasein's 'being-in-the-world'. Dasein is not a subject that stands apart from the world, but is embedded in the world. Heidegger argues that Dasein's being-in-the-world is characterized by a certain kind of understanding. Dasein understands itself and the world in terms of practical concerns and projects. This understanding is not a theoretical or abstract understanding, but a practical understanding. **Dasein's understanding** is always already oriented toward practical activities and projects. Heidegger calls this practical understanding 'being-in-the-world'. **Dasein's understanding** is not a static or fixed understanding, but is constantly changing and developing. Heidegger argues that **Dasein's understanding is shaped by** its **historical and cultural context**. Dasein is always already situated in a particular historical and cultural context, and this context shapes **Dasein's understanding of itself and the world**. Heidegger calls this historical and cultural context 'historicity'. Dasein's historicity is not just a matter of past events, but is a fundamental aspect of Dasein's being. Dasein is always already historical. Heidegger argues that **Dasein's understanding of itself and the world is shaped** by its **existential concerns**. Dasein is not just a passive recipient of its historical and cultural context, but is actively engaged in shaping its own **understanding of itself and the world**. This is what Heidegger means by Dasein's 'existentiality'. Dasein's existentiality is not just a matter of individual choice, but is a fundamental aspect of Dasein's being. Dasein is always already concerned with its own existence. Heidegger argues that **Dasein's understanding of itself and the world is shaped** by its **temporality**. Dasein is not just a present being, but is always already oriented toward the future and has a past. Dasein's temporality is not just a matter of time passing, but is a fundamental aspect of Dasein's being. Dasein is always already temporal. Heidegger calls this aspect of Dasein's being 'temporality'. Dasein's temporality is not just a matter of time passing, but is a fundamental aspect of Dasein's being. Dasein is always already temporal. Heidegger argues that **Dasein's understanding of itself and the world is shaped by** its **understanding of death**. Dasein is not just a biological organism, but is a being that is aware of its own death. Dasein's awareness of death is not just a fear of death, but is a fundamental aspect of Dasein's being. Dasein's awareness of death is what gives Dasein its authenticity. Heidegger calls this aspect of Dasein's being 'authenticity'. Dasein's authenticity is not just a matter of individual choice, but is a fundamental aspect of Dasein's being. Dasein is always already authentic. Heidegger argues that **Dasein's understanding of itself and the world is shaped by** its **understanding of anxiety**. Dasein is not just a being that is anxious about specific things, but is anxious about the very possibility of being.

(a)

··· For Heidegger, the world that happens is always a world of objects, a world of things, and it is also a world of persons. In this way, Dasein finds itself in a situation of **Being-in-a-world**, which is a situation in which Dasein has access to a temporal history. The Being of Dasein is **Being-in-a-world**, and this world is the Being-world of Dasein. But this **Being-world**, although it is a world of things, is also a world of persons, of a plurality of Daseins. **Being-in-a-world** is **Being-with**, and Dasein is always with other Daseins. This **Being-with** is the Being with others that Dasein has, and, in this way, Dasein finds itself in a situation of **Being-in-the-world**, where the world is a world of things and a world of persons. This **Being-in-the-world** is not simply **Being-with** others, however. This **Being-in-the-world** is also **Being-with-other-Daseins,** because Dasein is in the world with other Daseins. But Dasein is also in the world with things, because Daseins are also things. This B**eing-in-the-world**, then, is **Being-with-other-Daseins**, and it is also **Being-with-other-things**, because things are in the world with other Daseins. This **Being-in-the-world** is not simply Being-in-the-world-with-others, however. This **Being-in-the-world** is also **Being-in-the-world-with-others-themselves**, because others-themselves are in the world with other Daseins and with other things. And this **Being-in-the-world** is also **Being-in-the-world-with-others-themselves-themselves**, because **others-themselves-themselves** are in the world with other Daseins and with other things.

(b)

*Figure 6.* Examples of mode collapse that do not appear as explicit token loops: (a) template repetition with only a single word changed at each cycle; (b) conceptual looping without semantic progression, with superficial syntactic variation.

Looping typically manifests as explicit endless repetition, but it also includes "softer" forms of degeneration. Degeneration refers to poor-quality generations that are repetitive, incoherent, or bland (Holtzman et al., 2020). Figure 6 shows two examples of degeneration that are not manifested as explicit token loops. Figure 6(a) shows a template repetition with only a word replaced at each repetition. Figure 6(b) shows a conceptual looping without semantic progression, with meaningless syntactic variations.

## D. Efficient Estimation of the Regulated Subspace

This section describes how we efficiently estimate the regulated subspace $\mathbf{U} \in \mathbb{R}^{D \times c}$ (Eq. (10)) in high dimension. Directly forming $\mathbf{\Sigma}$ and $\mathbf{\Sigma}_\Delta$ and solving the generalized eigenvalue problem would be expensive at scale. Instead, we use a small number of iterations of a block power/orthogonal iteration scheme that only requires matrix–vector products and a QR-based orthonormalization step (Golub & van Loan, 2013; Saad, 2011).

**Problem setup.** Let $\mathbf{v}_t \in \mathbb{R}^D$ denote the (mean-centered) value-cache row at time $t$. Over a local window of length $T$, define

$$\mathbf{V}_0 = [\mathbf{v}_1, \ldots, \mathbf{v}_{T-1}]^\top \in \mathbb{R}^{(T-1) \times D}, \qquad \mathbf{V}_1 = [\mathbf{v}_2, \ldots, \mathbf{v}_T]^\top \in \mathbb{R}^{(T-1) \times D},$$

and optional time weights $\mathbf{w} \in \mathbb{R}^{T-1}$ with $\mathbf{W} = \mathrm{diag}(\mathbf{w})$ (we set $\mathbf{w} = \mathbf{1}$ if not specified). The generalized eigenvalue problem in Eq. (10) can be written in sample form as

$$\left( \tfrac{1}{2} \mathbf{V}_1^\top \mathbf{W} \mathbf{V}_0 + \tfrac{1}{2} \mathbf{V}_0^\top \mathbf{W} \mathbf{V}_1 \right) \mathbf{u} = \lambda \left( \mathbf{V}_0^\top \mathbf{W} \mathbf{V}_0 \right) \mathbf{u}, \tag{18}$$

up to normalization constants and a small diagonal regularizer.

**Direct solve vs. iterative approximation.** If solved directly, one would (i) form the $D \times D$ matrices in Eq. (18) at a cost of $O(TD^2)$ time and $O(D^2)$ memory, and (ii) compute the top-$c$ generalized eigenpairs at $O(D^3)$ time using a dense solver. In contrast, our estimator avoids forming $D \times D$ matrices and approximates the dominant $c$-dimensional eigenspace with a small number $K$ of block power/orthogonal-iteration steps. Each iteration uses only windowed projections and an orthonormalization, costing $O(TDc)$ time for projections plus $O(Dc^2)$ for orthonormalization, for a total complexity $O(K(TDc + Dc^2))$ (we use $c \leq 8$ and small $K$ in practice).

**Block power / orthogonal iteration.** We initialize $\mathbf{U}^{(0)} \in \mathbb{R}^{D \times c}$ randomly and orthonormalize it (QR). At iteration $k$, we form the windowed projections $\mathbf{P}_0 = \mathbf{V}_0 \mathbf{U}^{(k)}$ and $\mathbf{P}_1 = \mathbf{V}_1 \mathbf{U}^{(k)}$, and update

$$\widetilde{\mathbf{U}}^{(k+1)} = \mathbf{V}_0^\top \mathbf{W} \left( \mathbf{P}_1 \oslash (\mathbf{P}_0 + \varepsilon) \right), \qquad \mathbf{U}^{(k+1)} = \mathrm{orth}\left( \widetilde{\mathbf{U}}^{(k+1)} \right), \tag{19}$$

where $\oslash$ denotes elementwise division, $\varepsilon$ is a small constant for numerical stability, and $\mathrm{orth}(\cdot)$ denotes orthonormalization (implemented via QR decomposition).

**Generalized eigenvalue estimates.** After the final iteration, we estimate generalized eigenvalues using a Rayleigh-quotient form consistent with Eq. (10). Let $\mathbf{P}_0 = \mathbf{V}_0 \mathbf{U}$ and $\mathbf{P}_1 = \mathbf{V}_1 \mathbf{U}$. For each component $r \in \{1, \ldots, c\}$, we compute

$$\hat{\lambda}_r = \frac{\sum_{t=1}^{T-1} w_t (\mathbf{P}_1)_{t,r} (\mathbf{P}_0)_{t,r}}{\sum_{t=1}^{T-1} w_t (\mathbf{P}_0)_{t,r}^2 + \varepsilon}. \tag{20}$$

## E. Temporal Spectral Structure of Value-Cache

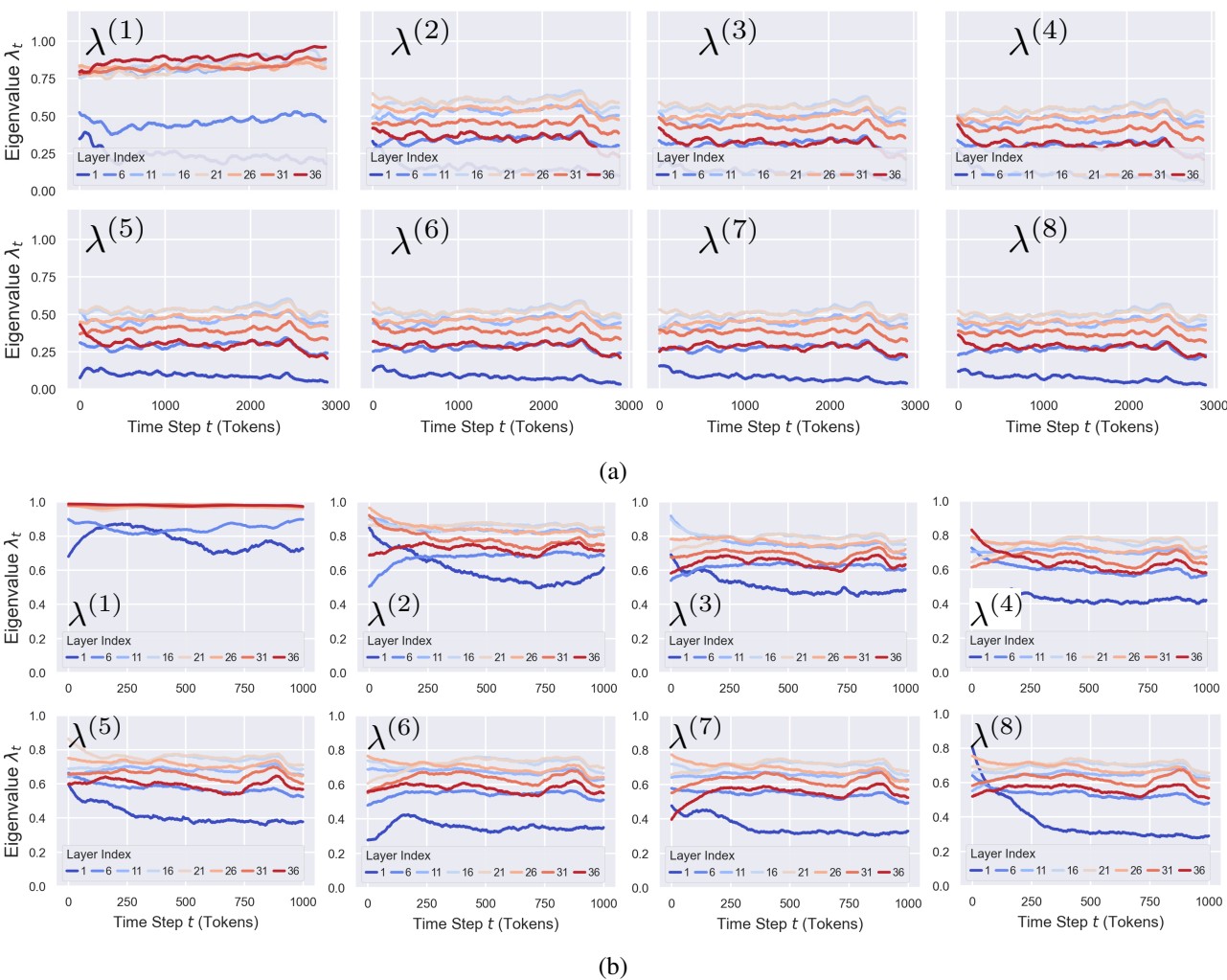

*Figure 7.* Evolution of the top-8 generalized eigenvalues over time in a non-collapse generation run (temperature $= 1.0$). (a) Mean eigenvalues over attention heads in each layer. (b) Maximum eigenvalues over attention heads in each layer.

Figure 7 shows the top-8 generalized eigenvalues of Eq. (10) across Transformer layers during a single *non-collapse* generation run with Qwen3-4B-Base, using the same prompt as in the main text (the first 1,000 tokens of HEIDEGGER from SEP) and temperature $= 1.0$. We use the standard filtering settings ($k = 50$, $p = 0.9$; Section 6) and apply *no intervention* (i.e., RMR is disabled). Eigenvalues are computed per attention head. We report both the mean over heads (Figure 7(a)), which reflects typical head-level persistence, and the maximum over heads (Figure 7(b)), which highlights the most persistent head in each layer at each time step.

Two salient patterns emerge. First, the dominant eigenvalue $\lambda^{(1)}$ stays close to 1 in most layers, indicating the presence of a highly persistent direction in the value cache even during normal generation. In contrast, layers closer to the input embedding (blue) exhibit smaller eigenvalues, consistent with these layers being more locally anchored to lexical/short-range structure and less dominated by long-term reinforcement. Second, the remaining eigenvalues typically fall below $0.8$ (often in the $0.5$–$0.8$ range) and are relatively stable over time in this non-collapse trajectory. This separation motivates our threshold choice $\lambda_{\min} = 0.8$: it is designed to selectively regulate only the most persistent direction(s) (often the leading eigenmode), while leaving the bulk of moderate-persistence components unchanged.

# F. Supplementary Results for Reinforced Mode Regulation

## F.1. Comprehensive Results on SEP Dataset

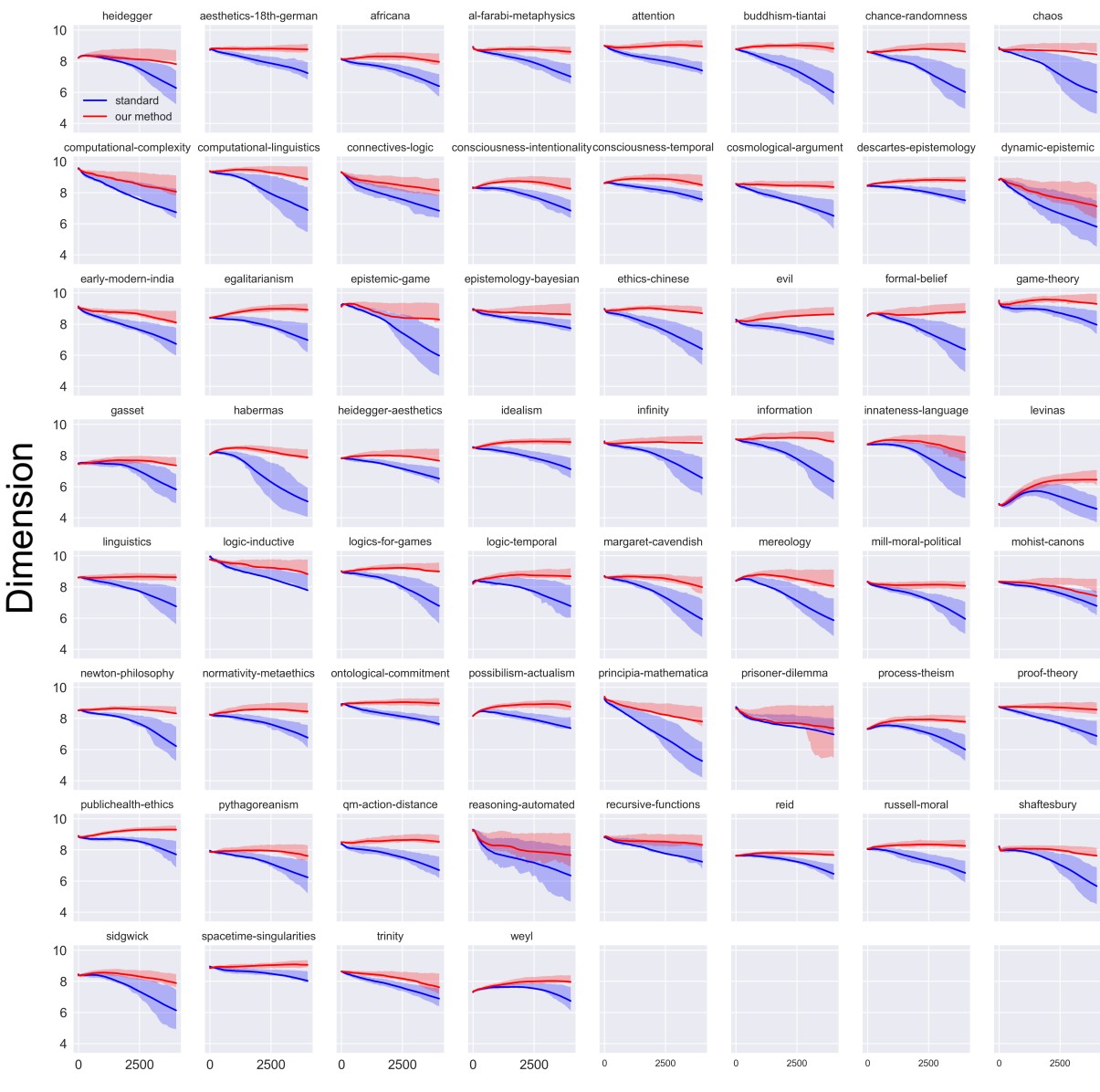

*Figure 8.* Results on 60 texts from the SEP dataset. Each pane shows the evolution of correlation dimension over 96 completions, without RMR (blue) and with RMR (red). Shaded areas indicate the 25% and 75% quantiles across completions.

Figure 8 shows the results on 60 texts selected from the SEP dataset. We generated 96 completions of 4,000 tokens for each text, and measured the correlation dimension at each timestep. Figure 8 shows the mean correlation dimension over all completions, and each pane shows the evolution of mean correlation dimension for a single text. Blue plots represent standard decoding and red plots represent RMR. The shaded areas indicate the 25% and 75% quantiles among different completions. The generation temperature was set to 1.0 throughout the generation process.

As seen, texts generated with RMR maintained stable correlation dimension even after 4,000 steps, significantly higher than standard decoding in most cases. This indicates that RMR is generally effective for text completion tasks.

## F.2. Other Language Models

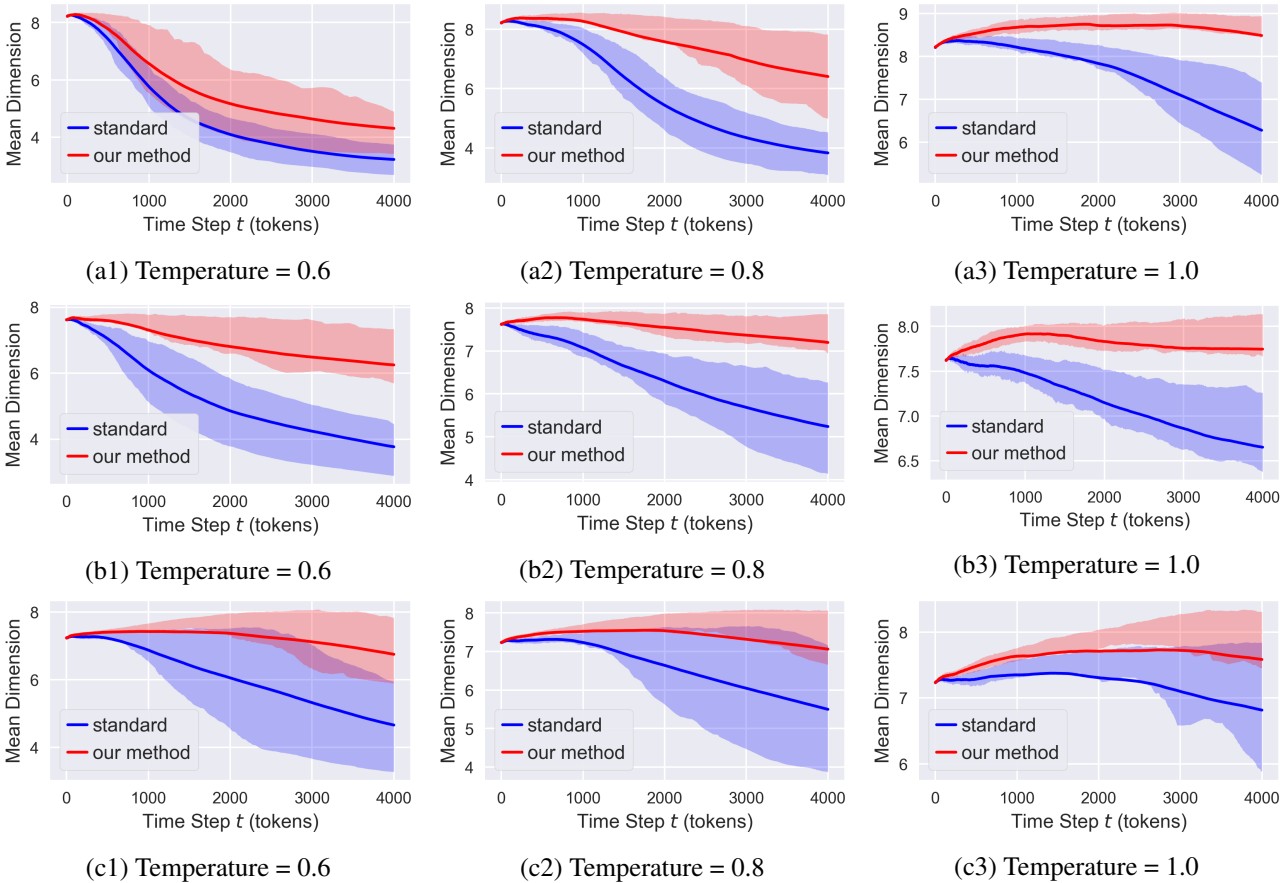

*Figure 9.* Correlation dimension trajectories for different LLMs completing HEIDEGGER at different temperatures. (a1–a3) Qwen3-4B-Base, (b1–b3) Qwen3-4B-Instruct, (c1–c3) Llama3.1-8B-Instruct.

Figure 9 shows the results of three different LLMs (Qwen3-4B-Base, Qwen3-4B-Instruct, Llama3.1-8B-Instruct (Grattafiori et al., 2024)) in completing the text HEIDEGGER at different temperatures. We generated 96 completions of 4,000 tokens for each LLM, and measured the correlation dimension at each timestep. Blue plots represent the standard decoding method and red plots represent RMR. The shaded areas indicate the 25% and 75% quantiles among different completions.

As in previous experiments, RMR generally maintains higher correlation dimension than standard decoding, indicating that RMR is applicable to different LLMs.

Instruction-tuned models (Qwen3-4B-Instruct and Llama3.1-8B-Instruct) showed slightly better resistance to collapse than the base model (Qwen3-4B-Base), with less drop in dimension value. When used with the instruction-tuned models, RMR still improves the collapse significantly.

# G. LLM-as-a-Judge for Text Evaluation

**LLM-as-a-Judge Prompt**

**Task.** You are given a FIXED CONTEXT text and a CONTINUATION generated by a language model.

**Your job.** Evaluate **only the quality of the generated continuation**, explicitly considering:

- how well it coheres with the given context;

- how it continues or breaks the structure;

- how much new information it contributes.

**Do not** evaluate truth, factual accuracy, or philosophical correctness.

**Input format (JSON).**

```
{
"context":  "<fixed context text>",
"generated":  "<LLM-generated continuation>"
}
```

**Important.** You score **only** the generated continuation, but all judgments must be **relative to the context**.

---

**Scoring system (0.0–10.0; floating-point allowed).**

---

Scores may be any real number from 0.0 to 10.0 (e.g., 2.3, 7.8, 9.4).
**Interpretation of ranges.**

- **9.0–10.0**: Exceptional quality; extremely rare; continuation is nearly seamless and strongly progressive.

- **7.0–8.9**: Solid, coherent, high-quality continuation with clear forward motion.

- **5.0–6.9**: Competent but shows moderate issues (redundancy, weak transitions, limited novelty).

- **3.0–4.9**: Noticeable problems; weak coherence, low progression, or style drift.

- **0.0–2.9**: Major failure modes; incoherence, breakdown, repetition loops, or degeneration.

---

**Evaluation dimensions (each 0.0–10.0).**

---

**Coherence & contextual consistency (0.0–10.0). Core question:** Does the continuation fit naturally with the context's topic, tone, abstraction level, and trajectory?

- **10.0**: Seamless continuation; fully consistent with contextual logic.

- **7.0**: Good alignment with minor mismatches.

- **5.0**: Partial connection; loosely attached to context.

- **3.0**: Clear mismatch or shift in focus/tone.

- **0.0**: Contextual breakdown or reset.

**Syntactic & stylistic control (0.0–10.0). Core question:** Is the prose tight, clear, and consistent with the context?

- **10.0**: Well-structured, controlled, stylistically aligned.

- **7.0**: Clear but somewhat dense or uneven.

- **5.0**: Heavy or awkward phrasing; minor grammar issues.

- **3.0**: Hard to parse or stylistically drifting.

- **0.0**: Degenerate or unreadable.

**Information progression / forward motion (0.0–10.0). Core question:** Does the continuation introduce new, meaningful content that advances the context?

- **10.0**: Strong advancement with clear conceptual progression.

- **7.0**: Good forward motion; some new claims but moderate depth.

- **5.0**: Mild progression; mostly elaboration.

- **3.0**: Minimal advancement.

- **0.0**: No progress; empty or circular.

**Output format (JSON only).**

Output a JSON object exactly in this format:

```
{
"scores": {
"coherence_contextual_consistency": <float 0.0--10.0>,
"syntactic_stylistic_control": <float 0.0--10.0>,
"information_progression": <float 0.0--10.0>
},
"justification": "<2--4 concise sentences explaining the core reasons for
the assigned scores.  Focus on qualitative distinctions, not numerical
restatement.>"
}
```

**Constraints.**

- No evaluation of factual correctness.

- Judgments must be relative to the context.

- Output must be valid JSON.

