# OpenReview forum: "Escaping Mode Collapse in LLM Generation via Geometric Regulation"
_ICML.cc/2026/Conference — ICML 2026 regular_

### Official Review · Reviewer_ExY7 · 2026-03-07

**Soundness:** 2
**Presentation:** 3
**Significance:** 3
**Originality:** 3
**Overall Recommendation:** 4
**Confidence:** 4

**Summary:**

This paper investigates mode collapse in autoregressive language model generation, where long decoding trajectories may converge to repetitive or low-diversity outputs. The authors propose interpreting this phenomenon as a geometric collapse of the generation trajectory, where the accessible region of state space shrinks to a low-dimensional regime. To quantify this effect, they introduce correlation dimension as a diagnostic metric computed from the trajectory of next-token distributions. Building on this perspective, the paper proposes Reinforced Mode Regulation (RMR), which detects temporally persistent directions in the transformer value cache using a generalized eigenvalue formulation and attenuates these directions during decoding. Experiments demonstrate that RMR significantly reduces collapse across temperature-locked and entropy-locked settings while preserving text quality relative to standard decoding methods. The paper thus aims to provide both a diagnostic framework and an intervention method for mitigating generation collapse.

**Compliance With Llm Reviewing Policy:**

Affirmed.

**Final Justification:**

My concerns about the external evaluation and toy model are largely addressed in the rebuttal, and I will keep my original rating.

**Key Questions For Authors:**

**Q1: External validation of collapse reduction**

The main metric for “non-collapse” is defined using the paper’s proposed correlation dimension. Could the authors provide additional validation using external metrics (e.g., repetition statistics, semantic redundancy measures, or human evaluation) to confirm that the improvements correspond to meaningful reductions in perceived degeneration?

**Q2: Robustness across prompts and domains**

Most main experiments are conducted with a single prompt (Heidegger text) and one primary base model. How consistent are the results across more diverse prompts (e.g., narrative, dialogue, code, reasoning tasks) and across additional models?

**Q3: Sensitivity to hyperparameters in RMR**

The method introduces several design choices (e.g., eigenvalue threshold and regulation strength 𝜂). How sensitive are the results to these hyperparameters?

**Q4: Clarification on Figure 2**

Fig 2 suggests a timing gap between different collapse signals. In panel (a), explicit looping appears to start very early (around 1050 tokens), whereas in panel (b) the earliest marked transition seems to occur much later (around 2500 tokens). In addition, the transition points of entropy and Distinct-2 are delayed relative to the drop in correlation dimension. What does the generated text typically look like during (i) the intermediate period after visible looping begins but before the sharp drop in correlation dimension, and (ii) the delay between the dimension transition and the later entropy/Distinct-2 transitions?

**Limitations:**

yes

**Strengths And Weaknesses:**

**Strengths**:
1. The work introduces a geometric perspective on generation collapse, framing it as a reduction in the effective dimensionality of the generation trajectory. I think this is an interesting and distinctive view. The use of the correlation dimension as a trajectory-level diagnostic and its connection to persistent modes in value-cache dynamics offer an interesting conceptual perspective beyond standard token-level decoding analysis.
2. The paper is generally well organized, and the high-level narrative is sound and easy to follow: it first proposes a geometric interpretation of collapse, then introduces correlation dimension as a diagnostic metric, and finally proposes RMR as an intervention. The experimental results, including entropy-locked and temperature-locked settings, provide consistent empirical support for this narrative.
3. Long-form degeneration and repetition remain a practical issue in LLM generation. A method that mitigates collapse at inference time without retraining the model could be useful for long-context generation scenarios and may motivate further work on state-space diagnostics for decoding dynamics.

**Weaknesses**:
1. The main evaluation metric for “non-collapse” is based on the paper’s own geometric diagnostic, which makes the empirical validation somewhat self-referential. Additional external collapse metrics or human evaluation would strengthen the claims.
2. The main experiments are relatively narrow in scope, focusing largely on one prompt and one primary base model in the main paper. Broader coverage across prompts, domains, and models would further validate the generality of the proposed method.
3. The state-dependent IFS in Section 3 is a useful intuition pump, but its connection to real transformer dynamics remains largely conceptual rather than theoretically established.

---

> ### Author Rebuttal · Authors · 2026-03-31
>
> **Thank you for the careful and detailed review.** We agree with the main concerns
> you raised and will revise the paper accordingly.
>
> ---
> ## External validation beyond correlation dimension
> > Could the authors provide additional validation using external metrics?
>
> We agree that additional evaluations beyond correlation dimension would
> strengthen the paper. The paper
> already includes exact looping rate, next-token entropy, Distinct-2,
> LLM-as-a-judge results and qualitative degeneration examples in the appendix.
> In the revision, **we will present these external metrics more clearly alongside
> the correlation-dimension results.** We will also add direct qualitative
> comparisons between standard decoding and RMR.
>
> ---
> ## Robustness across prompts and domains
> > How consistent are the results across more diverse prompts (e.g., narrative, dialogue, code, reasoning tasks) and across additional models?
>
> We agree that the main-text presentation is narrow. This was mainly for
> clarity. The appendix already includes broader evidence on 60 SEP texts and
> multiple LLMs, including Qwen3-4B-Instruct and Llama3.1-8B-Instruct.
>
> Following your suggestion, we also added an **instruction-following**
> experiment on **ConStory**. We randomly sampled 100 prompts where the model is
> asked to write an 8,000-10,000 token story. We again observed substantial mode
> collapse under standard decoding, including large drops in correlation
> dimension and long-range repetition, while RMR substantially mitigates the
> collapse. We will include a concise summary table in the revision.
>
> We agree that dialogue, code, reasoning, and RAG are also important settings.
> We will clarify that the current paper does not fully cover them, while making
> the broader evidence above more visible.
>
> ---
> ## Sensitivity to hyperparameters
> > How sensitive are the results to these hyperparameters?
>
> In practice, we find the method only weakly sensitive to the eigenvalue
> threshold. Because the slow directions in Eq. (10) have bounded eigenvalues
> ($|\lambda| \le 1$ under stationarity), the threshold has a stable meaning. In
> most healthy generations, the leading eigenvalues do not exceed $0.8$, so
> regulation is intermittent and becomes active only near collapse. This is why
> RMR has limited effect on overall text quality but a clearer effect on collapse
> mitigation.
>
> The regulation strength $\eta$ is also not very sensitive: it only needs to be
> set slightly below the thresholded eigenvalue level. In the paper we use
> $\eta=0.7$, i.e., slightly below $\lambda_{\min}=0.8$. Other hyperparameters,
> such as the decay rate $\gamma$ ($=0.995$), follow a similar logic. We will add
> more ablation results and discussion.
>
> ---
> ## Figure 2 and the timing gap
> > In panel (a), explicit looping appears to start very early (around 1050 tokens), whereas in panel (b) the earliest marked transition seems to occur much later (around 2500 tokens).
>
> We apologize for the confusion. The two panels illustrate different regimes of
> the same phenomenon. Figure 2(a) uses a **low-temperature** example to make
> explicit looping visually clear, so collapse appears early and sharply. Figure
> 2(b), by contrast, shows a more normal / **higher-temperature** regime, where
> geometric collapse can emerge but the transition to fully explicit looping may
> take much longer. Because of space limitations, this full long-horizon
> evolution is not visible in the main-text figure.
>
> > What does the generated text typically look like during (i) the intermediate period, and (ii) the delay?
>
> In the intermediate regime, the text typically already shows **degeneration,
> but not yet a fully explicit loop**: template repetition, semantic stagnation,
> or repeated local variations with superficial lexical change.
> We will clarify this in the caption and add
> more representative long-horizon examples in the appendix.
>
> ---
> ## Toy model
> > The state-dependent IFS in Section 3 is a useful intuition pump, but its connection to real transformer dynamics remains largely conceptual rather than theoretically established.
>
> We agree that its connection to real Transformer dynamics is conceptual rather
> than a formal derivation. The state-dependent IFS is intended only as a minimal
> dynamical model that isolates the feedback mechanism relevant to the
> long-horizon phenomenon, rather than as a faithful model of Transformer.
>
> Since our focus is on mode collapse and critical transition, our hope is that
> these simplifications mainly affect short-horizon details, while the qualitative
> long-horizon mechanism remains informative: nonlinear dynamical-systems theory
> is often most informative in predicting qualitative long-horizon phenomena such
> as critical transition. We will position this point more carefully in the
> revision.

---

> > ### Author Rebuttal · Reviewer_ExY7 · 2026-04-01
> >
> > Thank you for the detailed responses. The clarifications are helpful, and I will keep my original rating.

---

### Official Review · Reviewer_cB3K · 2026-03-12

**Soundness:** 3
**Presentation:** 3
**Significance:** 3
**Originality:** 3
**Overall Recommendation:** 4
**Confidence:** 2

**Summary:**

This work focuses on the mode collapse problem in the generation process of LLMs. It discusses the limitations of existing decoding approaches for addressing this problem. Then it tries to fill the research gap and proposes a new dynamical-systems perspective for investigating this issue. To support this view empirically, it conducts experiments using a newly designed metric (correlation dimension), which quantifies the degree of geometric collapse. Based on experimental analysis, this work introduces Reinforced Mode Regulation (RMR), a lightweight intervention method during inference to alleviate the collapse issue. Experiments across many LLMs and decoding settings demonstrate the effectiveness of RMR.

**Compliance With Llm Reviewing Policy:**

Affirmed.

**Final Justification:**

My concerns about the title, figure 2 presentation, and additional quantitative experiment are addressed in the rebuttal. So I keep my original rating.

**Key Questions For Authors:**

- (Just out of curiosity) Although this work focuses on the inference stage, can your methods, experiments, and analyses provide some insights for alleviating mode collapse problems in future LLM training?

**Limitations:**

Yes.

**Strengths And Weaknesses:**

**Strengths**
- The well-structured manuscript logically states the motivation and core idea.
- Many diagrams in the paper are clearly presented.
- The newly proposed dynamical-systems perspective, correlation dimension (metric to measure the geometric collapse), and Reinforced Mode Regulation method are novel and are presented with clear mathematical formulations.
- The authors conduct enough experiments to support their analysis and the effectiveness of their proposed method.

**Weaknesses**
- The title is too broad and fails to reflect the new view and the novel method proposed in this work.
- In Figure 2(a), the looping in generation starts at ~100 tokens. However, the turning point in Figure 2(b) occurs between 2000 and 3000 tokens. The inconsistency between these two is confusing.
- The manuscript provides quantitative experimental results to show the effectiveness of RMR. It’s recommended to include qualitative results (e.g., provide a concrete example and compare RMR with other methods).

---

> ### Author Rebuttal · Authors · 2026-03-31
>
> **Thank you very much for the positive and constructive review.** We appreciate
> your helpful suggestions and will revise the paper accordingly.
>
> ---
> ## Title is too broad
> > The title is too broad and fails to reflect the new view and the novel method proposed in this work.
>
> We agree that the current title is broader than the paper's actual
> contribution. In the revision, we will use a more specific title. It will more
> directly reflect both the geometric-collapse perspective and the proposed
> inference-time method. One possible revision is ``Escaping Mode Collapse in
> Long-Form LLM Generation **via Inference-Time Geometric Regulation**.''
>
>
> ---
> ## Figure 2(a) is confusing
> > In Figure 2(a), the looping in generation starts at ~100 tokens. However, the turning point in Figure 2(b) occurs between 2000 and 3000 tokens. The inconsistency between these two is confusing.
>
> We apologize for the confusion. The two panels are intended to illustrate
> different regimes of the same phenomenon. Figure 2(a) uses a **low-temperature**
> example to make explicit looping visually clear, so the collapse appears **early
> and sharply**. Figure 2(b), by contrast, is meant to show behavior in a more
> normal / **higher-temperature** regime. In that regime, geometric collapse can
> emerge, while the transition to fully explicit looping may take much
> longer. Because of space limitations, this full long-horizon evolution is not
> visible in the main-text figure. We will revise the caption and surrounding
> discussion to make this distinction clearer. We will also include additional
> long-horizon examples in the appendix.
>
>
> ---
> ## Include qualitative results
> > The manuscript provides quantitative experimental results to show the effectiveness of RMR. It’s recommended to include qualitative results (e.g., provide a concrete example and compare RMR with other methods).
>
> We agree that side-by-side qualitative examples would make the effect of RMR
> more intuitive. In the revision, we will add concrete generation examples under
> the same prompt. This will allow readers to compare standard decoding and RMR
> directly at the text level. It will also make it easier to inspect how RMR
> reduces explicit looping and softer forms of degeneration, such as template
> repetition and semantic stagnation.
>
> We will also provide a new illustration comparing the attention maps with
> regulation ON/OFF. RMR prevents the spectral gap from widening
> abnormally near collapse. This also helps avoid the abnormal concentration of
> attention weights. We agree that these text-level and mechanism-level
> comparisons would be more helpful here.
>
>
> ---
> ## Future training implications
> > Although this work focuses on the inference stage, can your methods, experiments, and analyses provide some insights for alleviating mode collapse problems in future LLM training?
>
>
> We think the analysis may suggest some possible training-time directions,
> although this is beyond the scope of the current paper. One possible direction
> is to use correlation dimension as a scoring signal to **construct training
> example pairs**, i.e., more desirable vs. less desirable generations, which
> could be useful for post-training. Another direction is to design **regularizers**
> based on geometric properties of the value-cache trajectory, for example to
> discourage overly persistent self-reinforcing directions. We will mention these
> more carefully as possible implications, rather than as direct claims of the
> paper. We thank you for this insightful question.

---

> > ### Author Rebuttal · Reviewer_cB3K · 2026-04-02
> >
> > Thank you for the detailed clarifications. I will keep my score.

---

### Official Review · Reviewer_dm3R · 2026-03-13

**Soundness:** 3
**Presentation:** 3
**Significance:** 2
**Originality:** 2
**Overall Recommendation:** 4
**Confidence:** 2

**Summary:**

This paper studies mode collapse in autoregressive LLM generation through a dynamical-systems lens. The main claim is that looping, repetition, and the loss of diversity are not merely token-level phenomena, but reflect a geometric collapse of internal generation trajectories in representation space, which in turn reduces state-space reachability. To diagnose this behavior, the paper adopts correlation dimension as a trajectory-level metric and introduces Reinforced Mode Regulation (RMR), an inference-time intervention that identifies and dampens the dominant persistent directions in the Transformer value cache. Empirically, the method substantially improves the non-collapse rate in low-randomness regimes, for instance from 8% to 56% at temperature 0.7 and from 5% to 33% at entropy 1.0. The paper further reports that, on generations that do not collapse, RMR does not noticeably harm surface-level text quality.

**Compliance With Llm Reviewing Policy:**

Affirmed.

**Final Justification:**

My concerns about the toy model and the limited breadth of evaluation were adequately addressed in the rebuttal, so I maintain my weak accept recommendation.

**Key Questions For Authors:**

Since the main efficacy metric in Figure 5 defines non-collapse using correlation dimension > 8, and the text-quality analysis is also restricted to samples selected by this criterion, could the authors provide additional evaluations that are independent of correlation dimension?

**Limitations:**

yes

**Strengths And Weaknesses:**

**Strengths**

1. The paper is well written overall. The motivation is clearly explained, and the proposed perspective and method are presented in a reasonably coherent and accessible manner.

2. The paper offers an interesting perspective by viewing degeneration in long-form LLM generation as a collapse in state-space reachability rather than simple token repetition.

**Weaknesses**

1. While the state-dependent IFS toy model is intuitively helpful, it remains rather far from the real dynamics of the Transformer value cache.

2.  The main-text results focus largely on SEP text continuation and the HEIDEGGER prompt, with little evidence on multi-turn dialogue, instruction-following, or RAG settings. Because mode collapse does not necessarily behave the same way across these scenarios, the current empirical support for generalization remains limited.


Overall, while I may not be an expert on mode collapse, I found the paper’s central idea interesting and thought-provoking. In particular, the attempt to reinterpret mode collapse through reduced state-space accessibility caused by geometric collapse feels potentially meaningful. That said, I think the paper would be strengthened by extending the experiments to a broader range of settings and by establishing a tighter theoretical link between the toy dynamical picture and the actual internal mechanisms of Transformers.

---

> ### Author Rebuttal · Authors · 2026-03-31
>
> **Thank you for the thoughtful and constructive review.**
> We are encouraged that you found the central perspective meaningful, and we agree with the main points you raised.
>
> ---
> ## Toy model
> > While the state-dependent IFS toy model is intuitively helpful, it remains rather far from the real dynamics of the Transformer value cache.
>
> We agree. The state-dependent IFS is meant as a minimal dynamical model, not a
> faithful model of Transformer KV-cache dynamics. Our intention was only to keep
> a minimal connection to Transformer-like state evolution, in a highly restricted
> setting, while isolating the feedback mechanism relevant to the long-horizon
> phenomenon studied here. Since our focus is on accessibility collapse and
> critical transition, our hope is that these simplifications mainly affect
> **short-horizon** details, while the **qualitative long-horizon** mechanism of
> collapse and critical transition remains informative. We will position this
> point more carefully in the revision.
>
> ---
> ## Broader evaluation in more scenarios
> > The main-text results focus largely on SEP text continuation and the HEIDEGGER prompt, with little evidence on multi-turn dialogue, instruction-following, or RAG settings.
>
> We agree that the main-text presentation is narrow. This was mainly for
> clarity: we fixed the prompt/model to keep the geometric narrative easy to
> follow. The appendix already includes broader evidence on 60 SEP texts and
> multiple LLMs, including Qwen3-4B-Instruct and Llama3.1-8B-Instruct.
>
> Following your suggestion, we have also added an instruction-following
> experiment on the ConStory benchmark. We randomly sampled 100 prompts in the
> setting where the model is asked to write an 8,000-10,000 token story. **We
> again observed substantial mode collapse under standard decoding, including
> abrupt drops in correlation dimension and long-range repetition, while RMR
> substantially mitigates the collapse.** We will include a concise summary table
> in the revision.
>
> We agree that multi-turn dialogue and RAG are also important settings, and we
> will clarify that the current paper does not fully cover them.
>
> ---
> ## Additional evaluations independent of correlation dimension
> > Since the main efficacy metric in Figure 5 defines non-collapse using correlation dimension > 8, and the text-quality analysis is also restricted to samples selected by this criterion, could the authors provide additional evaluations that are independent of correlation dimension?
>
> We agree that additional evaluations independent of correlation dimension would
> strengthen the paper. Our intention was to use correlation dimension as a
> trajectory-level diagnostic, not as the sole notion of quality. The paper
> already includes exact looping rate, next-token entropy, Distinct-2, and
> LLM-as-a-judge results, and in the revision **we will present these more clearly
> alongside the correlation-dimension results**. We will also add a direct
> qualitative comparison between standard decoding and RMR.
>
> ---
>
> In short, we will revise the paper to clarify the role of the toy model, make
> the broader empirical evidence more visible, and include the new ConStory
> instruction-following results together with clearer external validations.

---

> > ### Author Rebuttal · Reviewer_dm3R · 2026-04-03
> >
> > Thank you to the authors for the response. I appreciate the clarifications provided, and I will maintain my positive score.

---

### Decision · Program_Chairs · 2026-04-30

**Decision:**

Accept (regular)

**Comment:**

This work reinterprets mode collapse in autoregressive LLM generation as geometric collapse of the internal generation trajectory. The authors introduce correlation dimension as a trajectory-level diagnostic and propose to dampen the dominant persistent directions during the inference. All three reviewers found the central perspective interesting and distinctive. The interpretation, diagnostic and the empirical method were also appreciated by reviewers. Concerns regarding the scope of evaluation, and connections to the realistic transformer settings were raised and addressed with additional experiments. The authors are encouraged to take all new results and discussions in the revision.